# Preparation of Cemented Carbide Insert Cutting Edge by Flexible Fiber-Assisted Shear Thickening Polishing Method

**DOI:** 10.3390/mi13101631

**Published:** 2022-09-29

**Authors:** Lanying Shao, Yu Zhou, Wei Fang, Jiahuan Wang, Xu Wang, Qianfa Deng, Binghai Lyu

**Affiliations:** 1College of Mechanical Engineering, Zhejiang University of Technology, Hangzhou 310023, China; 2Key Laboratory of Special Purpose Equipment and Advanced Processing Technology, Ministry of Education and Zhejiang Province, Zhejiang University of Technology, Hangzhou 310023, China

**Keywords:** shear thickening polishing, flexible fiber, cutting edge preparation, polishing angle, polishing speed, cutting edge radius

## Abstract

Reasonable cutting edge preparation can eliminate microscopic defects and improve the performance of a cutting tool. The flexible fiber-assisted shear thickening polishing method was used for the preparation of cemented carbide insert cutting edge. The influences of the polishing angle and polishing speed on the cutting edge preparation process were investigated, and the cutting edge radius and *K*-factor were employed as evaluation indexes to evaluate the edge shape. A prediction model of the cutting edge radius was also established using the mathematical regression method. The results show that the polishing angle has a more significant effect on the cutting edge radius. The cutting edge preparation efficiency is the highest under the polishing angle of 10°, and the cutting edge radius increased from the 15 ± 2 μm to 110 ± 5 μm in 5 min. The cutting edge shape can be controlled by adjusting the polishing angle, and the *K*-factor varies from 0.14 ± 0.03 to 0.56 ± 0.05 under the polishing angle (from −20° to 20°). The polishing speed has a less effect on the cutting edge radius and shape, but increasing the polishing speed within a certain range can improve the efficiency of cutting edge preparation. The flank face roughness decreased from the initial *Ra* 163.1 ± 10 nm to *Ra* 5.2 ± 2 nm at the polishing angle of −20°, which is the best polishing angle for the flank face surface roughness. The ANOVA method was employed to evaluate the effective weight of the polishing angle and polishing speed on preparation efficiency. The polishing angle (86.79%) has the more significant influence than polishing speed (13.21%) on the cutting edge preparation efficiency. The mathematical regression method was used to establish the model of the prediction of the cutting edge radius with polishing angle and speed, and the models were proved rationally. The results indicate that the FF-STP is an effective method for the high consistency preparation of cemented carbide insert cutting edge.

## 1. Introduction

Cemented carbides have the advantages of high hardness and being abrasion resistant, which allows them the largest application in the manufacturing of cutting tools and wear parts [1]. The cutting edge of a cemented carbide insert is mostly formed by grinding [2]. During the grinding process, defects, such as burrs, nicks, and thermal deformation, are unavoidably produced and seriously decline the performance and life of the cutting tools. The shape of the cutting edge greatly affects the magnitude and distribution of the thermal and mechanical load during material removal [3]. Cutting edge preparation is a technique that creates a defined shape and size of the cutting edge, and the aim of cutting edge preparation is to increase the strength of the cutting edge, reduce the internal stress of the coating, reduce the risk of edge chipping, and prolong the tool life [4].

Much research has been carried on the preparation of the cutting edge of cemented carbide. As one of the most commonly used methods for cutting edge preparation, Lv et al. [5] used the drag finishing (DF) to prepare the solid carbide end mills. The experimental results demonstrate that the cutting edge shape K-factor, MRR, and surface morphology are affected by the abrasive particle size, composition, and processing time. Zhao et al. [6] established a simulation model of the tool edge preparation process of the drag finishing (DF) method through the discrete element software EDEM and provided a basis for edge preparation optimization. Zlamal et al. [7] used abrasive jet machining and drag finishing to prepare cutting edges for PVD coated cutting tools, and this method can reduce the surface roughness of the tool and improve the surface integrity of deposited layers of PVD coating. Voina et al. [8] used wet blasting to prepare cutting edge rounding, and through the hole expansion test. It was found that the prepared cutting edge had a significant improvement in wear resistance and metal coating adhesion. Vopát et al. [9] proposed a novel method called edge preparation by plasma discharges in electrolyte (PDE), and the cutting edge radii were formed by immersing the cutting tool into an electrolyte. Laser machining is also used in the preparation of tool edges, and the parameters used for laser machining have a significant impact on the machined surface, surface integrity, and tool performance [10]. Vopát et al. [11] found that the cemented carbide turning inserts with larger cutting edge radius were worn out faster during the machining and the tool life increased when the cutting edge radius was smaller. DenKena et al. [12] proposed a novel method for the preparation of the cutting edge rounding using flexible bond diamond polishing tools, where the polishing tool can produce both asymmetrical rounding and symmetrical cutting edge rounding with a high repeatability. Bergs et al. [13] used diamond-studded brushing tools for the preparation of the cutting edges of carbide tools and produced symmetrical and asymmetrical cutting edges in combination with various material removal rates. Although the above method can play a good role in cutting edge preparation, a new cutting edge preparation process with high efficiency, fine surface quality, and controllable cutting edge shape is always expected.

Shear thickening polishing (STP) is a new ultra-precision machining method utilizing the rheological properties of non-Newtonian fluid under shear stress to achieve surface polishing. The method has good adaptability to complex surfaces and can achieve efficient and high-quality polishing [14]. Lyu [15] et al. conducted experiments on brush tool-assisted shear thickening polishing for cemented carbide insert, where the average surface roughness *Ra* of the seven measuring points on the cutting edge decreased from 118.01 to 8.13 nm in 10 min polishing, while the standard deviation decreased from 7.40 to only 0.3 nm. Chan [16] et al. showed that asymmetric and symmetrical cutting edges can be produced by honing tool edges using non-Newtonian fluids.

The flexible fiber-assisted shear thickening polishing (FF-STP) method was initially applied to polishing cemented carbide inserts with excellent effect. In this study, the FF-STP method was applied to the cutting edge preparation to explore the feasibility of the method and study the influence of polishing angle, polishing speed on the cutting edge forming, and obtain the shape change law under different polishing method parameters and the cutting edge radius change prediction model.

## 2. Principle of FF-STP

The FF-STP principle of cemented carbide insert cutting edge is shown in Figure 1. The polishing slurry is prepared by mixing abrasive particles with non-Newtonian fluid and additives. Under the applied shear stress, when the relative velocity exceeds a threshold, the shear thickening phenomenon occurs in the contact area between the workpiece and the polishing slurry. When the polishing tank rotates, the polishing liquid and the insert move relative to each other, and the multi-hydroxyl polymer dispersed in the polishing liquid aggregates into a large number of particle clusters under the action of shear stress, wrapping the abrasive particles, and forming “flexible fixation abrasives” that fit the insert and remove the material on the surface of the insert. However, because of its inertia effect, the polishing slurry flows along the rake face after thickening on the rake face and polishes the rake face rather than removing the material at the tip of the insert. So, it is not easy to form a relative motion with the cutting edge, as it cannot achieve a preparation effect on the cutting edge effectively. Therefore, the flexible fiber-assisted shear thickening polishing is introduced. The flexible fibers are able to apply elastic force to the polishing slurry and push the polishing slurry to the cutting edge to increase its material removal rate so as to improve the preparation efficiency.

## 3. Experimental Process and Conditions

### 3.1. Experimental Process

Figure 2 presents the experimental device. The workpiece is installed on a fixture which can adjust fixing angle, and the polishing slurry flows through the rake face frontally. The polishing angle α is defined as the angle between the rake face and the vertical direction. The angle between the flow direction of polishing slurry and the rake face is 90-α. The diameter of the polishing tank is 400 mm, and the polishing slurry is evenly and adequately attached to the surface of the flexible fibers. The flexible fibers fixed in the polishing tank drives the polishing slurry to rotate, and the insert remains stationary. When the relative velocity between the slurry and the insert exceeds the critical value, the shear thickening effect of the slurry is trigged, and the abrasive particles in the polishing slurry produce micro-cutting on the workpiece surface under the shear thickening effect and the assistance of flexible fibers.

### 3.2. Experimental Conditions

Figure 3 shows the carbide insert used in this study, the diameter of the insert basic circle is 25 mm, the length of the cutting edge is 7 mm, the rake angle is 0°, and the flank angle is 7°. The cutting edge radius is 15 ± 2 μm (Avg. ± Std.). Table 1 shows the mechanical properties of cemented carbide insert.

Figure 3b shows the measurement position of the cutting edge preparation. In order to ensure the reliability of the measurement, the workpiece surface was divided into three areas: A, B, and C. The cutting edge radius on 100 points in each area was measured, and the average cutting edge radius value of 300 points was calculated as the final result. The radius was measured by Zoller-pomSkpGo (Zoller Shanghai, Ltd, Shanghai, China) cutting edge measuring instrument, and the surface morphology of the insert was observed by the scanning electron microscope (SEM) (Sigma300, Zeiss, Germany). The roughness of three different positions on the machined surface were measured by the white light interferometer (Super View W1, Chotest, Shenzhen, China). The three measurement results are averaged, and the positions of the measurement points are shown in Figure 3c.

The experimental conditions are listed in Table 2. According to the previous study [15], the polishing slurry with a 6 wt.% concentration of 5000# diamond abrasive was prepared, and the flexible fiber tool used in the experiment is made of pig bristles with a fiber height of 30 mm, a fiber diameter of 200–250 μm, and a fiber density of 200–250 pieces/cm^2^. The cutting edge is immersed in the flexible fiber with a depth of 2 mm and is located 2 mm from the side of the polishing tank. The first group of the experiments was to explore the change rule of the cutting edge radius and shape under different polishing angles with the increase of processing time in 5 min. The polishing angle with the highest preparation efficiency was selected and applied to the second group of experiments. In the second group of experiments, the influence of polishing speed on the cutting edge radius and shape were explored. The polish speed is expressed by the rotation speed of the polishing tank in this study.

The third group of experiments was to explore the degree of influence of the polishing angle and polishing speed on the cutting edge radius. The Taguchi method was used to design an orthogonal experiment, and then the experimental results were converted into a signal-to-noise (*S/N*) ratio. By analyzing the *S/N*, the optimal polishing parameters were obtained, and the variance analysis of the *S/N* ratio was carried out to obtain the influence of different factors on the cutting edge radius. Three groups of parameters with the best results in the first and second group of experimental results were selected for the experiment, which are the polishing angles 0°, 10°, and 20° and polishing speeds of 60 r·min^−1^, 70 r·min^−1^, and 80 r·min^−1^ and the experimental scheme is shown in Table 3. The processing time for each trial was 2 min.

The polishing slurry was prepared by micro polyhydroxy polymer powders (Shandong, Hengren), deionized water, diamond particles (Zhengzhou Institute of Abrasives and Grinding, Zhengzhou, China), preservative (Casson), and dispersant (Ningbo Risheng New Material Co., Ltd, Ningbo, China). Figure 4 presents the rheological curves of the prepared slurry, which were measured by the stress-controlled rheometer (MCR 302 Anton Paar) under a temperature of 25 °C. The rheological curve of slurry displays four different shear–rheology zones: Zone I shows the slight shear thickening under a low shear rate, Zone II appears slightly shear thinning with the increase of shear rate from 0.3 s^−1^ to 3 s^−1^, Zone III presents strong shear thickening in the range of shear rate 3 s^−1^ to 100 s^−1^, and Zone IV shows strong shear thinning as the shear rate exceeds 100 s^−1^.

## 4. Results and Discussion

### 4.1. Comparative Experiment of STP and FF-STP on Cutting Edge Preparation

In order to investigate the preparation effect of shear-thickening polishing, the insert was processed for 5 min by the STP and the FF-STP at the polishing angle of 10° and the rotating speed of 70 r·min^−1^. It was found that the cutting edge radius of the edge increased from 15 ± 2 μm to 16 ± 2 μm by STP, while it increased from 15 ± 2 μm to 110 ± 5 μm by FF-STP (shown in Figure 5). The surface topography observed by white light interferometer was shown in Figure 6. The surface roughness of flank face by FF-STP is slightly lower than that by STP. It can be seen that the STP has no obvious effect on the edge preparation, the main reason is that most of the “flexible fixed abrasives” continue to move forward along the fluid after cutting and sliding on the rake or flank face because of its inertia effect, so it cannot continue to remove the surface material at the tip of the insert, then it is difficult to meet the cutting edge preparation requirements in a short time. However, with the assistance of flexible fibers, “flexible fixed abrasives” can be pushed to the surface of the cutting edge to continue cutting and sliding the cutting edge. In addition, during the contact process, the flexible fiber will exert certain pressure on the polishing slurry contacting the tool, so that it has a better shear-thickening effect to achieve cutting edge preparation in a short time.

### 4.2. Influence of Polishing Angle

To study the influence of different polishing angles on the preparation effect of cemented carbide insert edge by FF-STP, experiments were carried out under five different polishing angles at the speed of 70 r·min^−1^. The change of cutting edge radius with processing time is shown in Figure 7.

It can be seen that the cutting edge preparation efficiency is the highest when the polishing angle is 10°, and the cutting edge radius increases from 17 μm to 110 μm after 5 mins’ polishing. While the cutting edge preparation efficiency is the lowest when the polishing angle is −20° and the cutting edge radius increased from 15 μm to 37 μm only. From the overall trend, the cutting edge preparation efficiency in the first 2 min is higher than that in following 3 min. After 5 mins of polishing, the shape and the cutting edge radius change under different polishing angles are presented in Figure 8. When the polishing angle is −10° and −20°, the shape of the cutting edge changes obviously due to the change of the flank surface. When the polishing angle is larger than 0°, the shape of the cutting edge changes due to the rake face. The initial value of the *K*-factor is 0.14, *K*= *s_α_/s_γ_*, *S*_α_ is cutting edge segment on flank face, *S*_γ_ is cutting edge segment on rake face [17], and the *K*-factor changes with the polishing angle, as shown in Figure 9. The initial cutting edge shape is shown in Figure 10a. With the increase of the angle (from −20° to 20°), the value of *K* shows a trend of continuous decrease. At the polishing angle of −20°, the flank face is mainly polished, so *s_α_* increases faster than *s_γ_* (Figure 10b), and the ratio of *s_α_/s_γ_* reaches the maximum value. As the polishing angle continues to increase, the material removal efficiency on the rake face increases and the polishing angle reaches 20°, where the *K*-factor reaches a minimum (Figure 10c). The cutting edge shape can be controlled by adjusting the polishing angle.

The schematic illustration of cutting edge preparation at different polishing angles is shown in Figure 11, where *v*_1_ and *v*_2_ are the flow velocity of the polishing slurry along the flank and rake face, respectively. When the polishing angle is −20°, the polishing slurry almost impacts the edge of the insert and the relative speed between the polishing slurry and the edge decreases. As soon as the polishing slurry reaches the cutting edge, it divides into two streams, one of which flows to the rake face, *v*_2_ gradually decreases to 0 due to the action of friction and gravity. Another slurry flows to the flank face to polish the flank face with the flexible fiber. When the polishing angle is −10°, the relative motion rate of the polishing slurry and the cutting edge increases, and the material removal rate at the tip of the insert increases. When the polishing angle is 0°, the polishing slurry converges and stays on the rake face, but part of the polishing slurry also can form a good relative motion with the cutting edge. The flow velocity *v*_1_ is large, so the material removal rate continues to rise. When the polishing angle is 10°, the polishing slurry stays on the rake face and then flows to the flank face. The relative rate between the polishing slurry and the cutting edge is large, the pressure on the cutting edge increases, and the material removal rate increases. When the polishing angle is 20°, the polishing slurry flows smoothly through the rake face with the flexible fiber. However, the pressure on the cutting edge is reduced because the polishing slurry that stays on the rake face is reduced and the material removal rate decreases. The initial roughness *Ra* of the flank face is 163.1 nm. The polishing effect of the flank face at different polishing angles is shown in Figure 12. The polishing effect is the best at −20° (the roughness *Ra* of the flank face after polishing is 5.2 nm) and the worst at 20° (the roughness *Ra* value of the flank face after polishing is 143.3 nm). The polishing slurry has a good material removal effect on the rake face under the polishing angle 20°, while the polishing effect is not obvious at other angles (Figure 13). The flank and rake face of the cemented carbide insert after polishing are shown in Figure 14.

### 4.3. Influence of Polishing Speed

In order to explore the influence of polishing speed on the cutting edge preparation effect, the polishing angle with the highest cutting edge preparation efficiency was selected as 10°, and the experiments were carried out at five polishing speeds. Figure 15 shows the change of the cutting edge radius with processing time.

With the increase of polishing time, the cutting edge radius increases continuously. When the polishing speed increases from 50 r·min^−1^ to 70 r·min^−1^, the cutting edge preparation efficiency increases, and when the polishing speed is greater than 70 r·min^−1^, the efficiency of the cutting edge preparation decreases. This is because the shear rate of the polishing slurry is positively correlated with the polishing speed. Within a certain range of shear rate, the viscosity of the polishing slurry increases with the increase of shear rate, the force on the abrasive grains applied by the polishing slurry is enhanced, thus it can improve the cutting edge material removal efficiency. While the polishing speed is too fast, the abrasive particles are thrown to the polishing tank due to centrifugation [18], the mass fraction of abrasive particles involved in cutting edge material removal will decrease, and the removal efficiency of cutting edge material will decrease.

Figure 16 shows the changes of the cutting edge shape and radius at different polishing speeds. The basic cutting edge shape does not change very much at the speeds of 50 r·min^−1^ and 60 r·min^−1^. Then, with the increase of the polishing speed, the cutting edge approaches a symmetrical cutting edge. The *K*-factor changes with the polishing speed, as shown in Figure 17. With the increase in polishing speed, *K*-factor first increases and then essentially remains unchanged. The reason is that under the condition of 10°, the rank and flank face are both polished; however, the removal of the rake face is slightly larger than that of the flank face. With the increase of speed, the edge *s_α_*/*s_γ_* gradually increases. It can be seen that the polishing speed has less effect on the edge shape, but increasing the polishing speed within a certain range can improve the cutting edge preparation efficiency.

### 4.4. Analysis of Orthogonal Experiment Results

In order to explore the degree of influence of the polishing angle and speed on the cut-ting edge radius and analyze the interaction between polishing angle and polishing speed, an orthogonal experiment based on Taguchi method was used. Usually, there are three categories of quality characteristic in the analysis of the *S*/*N* ratio, including the-lower-the-better, the-higher-the-better, and the nominal-the-better [19], and the cutting edge radius belongs to the-higher-the-better. Equation (1) is used to calculate the *S/N* ratio.
(1)$$S/N=-10\log\frac{1}{r}\overset{r}{\underset{j=1}{\sum }}\frac{1}{H^{2}}$$
where *j* is the trial number, *r* is the total number of samples, and *H* is the value of the cutting edge radius.

The material removal of the cutting edge mainly depends on the relative velocity between the polishing slurry and the cutting edge surface and the applied pressure. When the polishing angle is too large, the polishing slurry flows smoothly through the rake face with the flexible fiber but the pressure on the cutting edge is reduced. When the polishing angle is too large, the polishing slurry almost impacts the edge of the insert and the relative speed between the polishing slurry and the edge decreases. Changing the polishing speed in these two cases can hardly change the material removal rate efficiently at the cutting edge. In the first group of experiments, the angles 0, 10, and 20 and the polishing speed 60 r·min^−1^, 70 r·min^−1^, and 80 r·min^−1^ with higher preparation efficiency were selected. The results of *R* and *S/N* are shown in Table 4. With the increase of the polishing speed, the preparation efficiency first increases and then decreases. With the increase of the polishing angle, the preparation efficiency first increases and then decreases. Figure 18 shows the variation of *R* with the average response of *S/N*, and the best solution for improving cutting edge preparation efficiency is the polishing angle of 10 and the polishing speed of 70 r·min^−1^. Figure 19 shows the ANOVA results for *R*. Polishing angle (86.79%) has the more significant influence, and polishing speed (13.21%) has the less influence on the cutting edge preparation efficiency.

The cutting edge prepared under the optimal conditions (polishing angle 10°, polishing speed 70 r·min^−1^) was observed and compared with the original cutting edge. As shown in Figure 20, the original cutting edge has a large number of serrated micro-notches, and there are many scratches of different depths on the surface. After the cutting edge is prepared, the burrs and scratches are completely removed, and the cutting edge is smooth.

## 5. Cutting Edge Radius Model

The prediction model of cutting edge radius was established by mathematical regression method [20,21]:

Model 1: Taking the polishing angle and polishing time in the cutting edge preparation parameters as independent variables and the radius of the cutting edge as dependent variables, mathematical regression analysis was conducted according to the experimental results to obtain the prediction model of the cutting edge radius as follows, and the fitting surface diagram of model 1 is shown in Figure 21:*Z* = 28.04563 + 79.1369 exp(−0.5((*T*−5.5281)/2.5457)^2^ − 0.5((*A* − 6.33336)/10.38034)^2^)(2)
where *Y*_1_ is the cutting edge radius (μm), *A* is the polishing angle (r·min^−1^), and *T* is the polishing time (min).

Model 2: Taking the polishing speed and polishing time in the cutting edge preparation parameters as independent variables and the radius of the cutting edge as dependent variables, mathematical regression analysis was conducted according to the experimental results to obtain the prediction model of the cutting edge radius as follows, and the fitting surface diagram of model 2 is shown in Figure 22:*Z* = −2.92836 + 128.1582 exp(−0.5((*T* − 7.38934)/4.489)^2^ − 0.5((*B* − 77.06983)/21.20732)^2^)(3)
where *Y*_2_ is the cutting edge radius (μm), *B* is the polishing speed (r·min^−1^), and *T* is the polishing time (min).

The weighted least squares optimization algorithm was used to correct the proposed model (Equations (2) and (3)), and the results are shown in Table 5 and Table 6. The *R*-square of Equations (2) and (3) were 0.93815 and 0.94947, respectively, which proved the rationality of the established model.

## 6. Conclusions

The flexible fiber-assisted shear-thickening polishing (FF-STP) method was used for the cutting edge preparation of cemented carbide insert, the effects of the polishing angle and the polishing speed on the forming of the edge were studied. According to the above experimental analysis, important conclusions have been drawn as follows:The efficiency of the cutting edge preparation by FF-STP is higher than STP. The flexible fibers play the role of pushing the polishing slurry to the cutting edge to increase its material removal rate, which achieves cutting edge preparation effectively.The polishing angle has a more significant effect on the cutting edge radius during the process. The cutting edge preparation efficiency is the highest under the polishing angle of 10°, where the cutting edge radius increased from the 15 ± 2 μm to 110 ± 5 μm in 5 min. The cutting edge shape can be controlled by adjusting the polishing angle, and *K*-factor varies from 0.14 ± 0.03 to 0.56 ± 0.05. The polishing speed has a less effect on the cutting edge radius and shape, but increasing the polishing speed within a certain range can improve the efficiency of cutting edge preparation.Two models that the cutting edge radius can be predicted as a function of polishing angle and polishing speed are established and the *R*-square of the two equations were 0.93815 and 0.94947, respectively, which proved the rationality of the established model.FF-STP can not only achieve the controllable change of the cutting edge radius and shape but also have a significant impact on the improvement of the insert surface quality. In the future, extensive research is required to clarify the material removal mechanism of the cutting edge during FF-STP processing.

## Figures and Tables

**Figure 1 micromachines-13-01631-f001:**
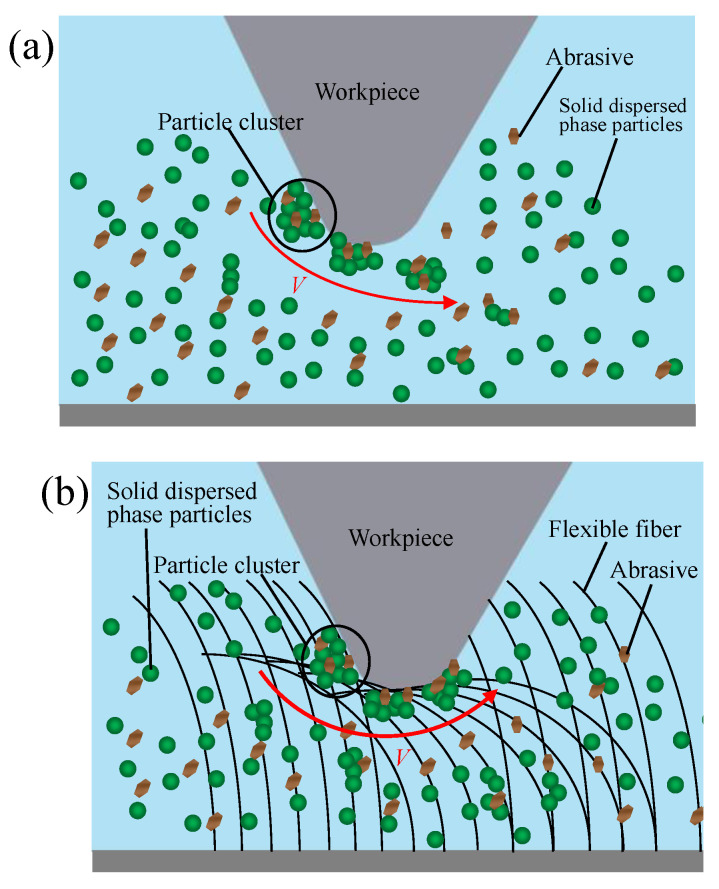
Schematic illustration of FF-FRP process for cemented carbide insert preparation: (**a**) STP, (**b**) FF-STP.

**Figure 2 micromachines-13-01631-f002:**
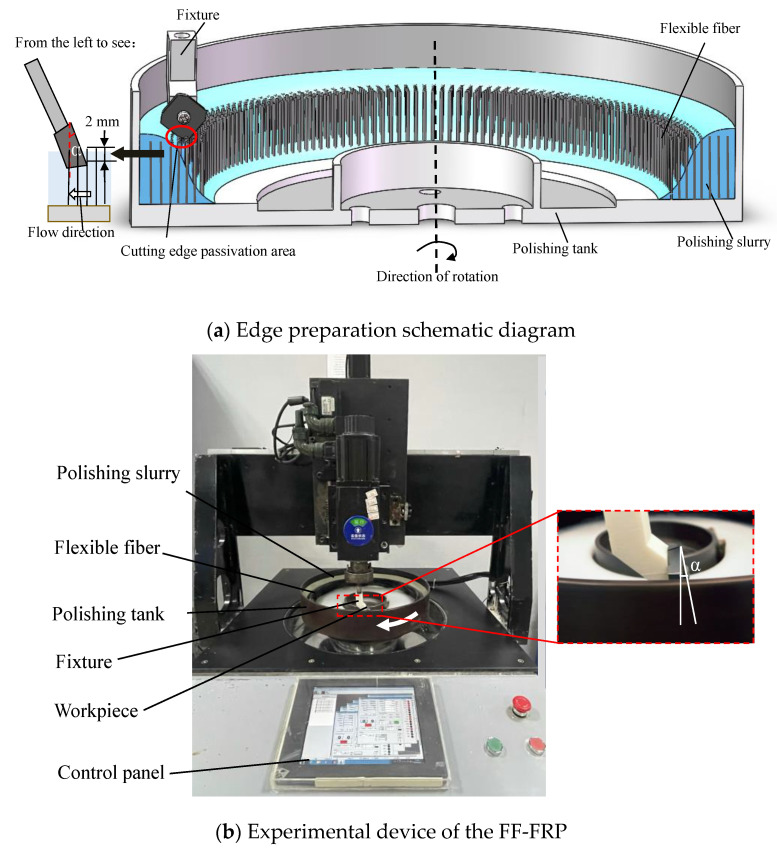
The FF-FRP experimental device (**a**) Schematic illustration. (**b**) Picture of experimental device.

**Figure 3 micromachines-13-01631-f003:**
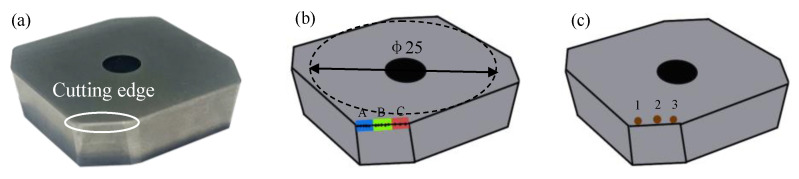
The cemented carbide insert: (**a**) The schematic diagram of cutting edge, (**b**) the measured position of the cutting edge radius, (**c**) the measured position of the cutting edge roughness.

**Figure 4 micromachines-13-01631-f004:**
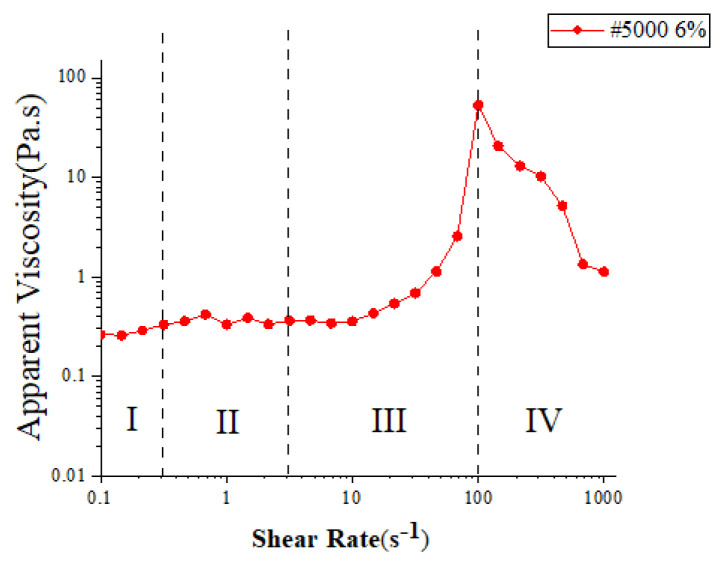
Rheological curves of 5000# diamond STP slurry at a concentration of 6 wt.%.

**Figure 5 micromachines-13-01631-f005:**
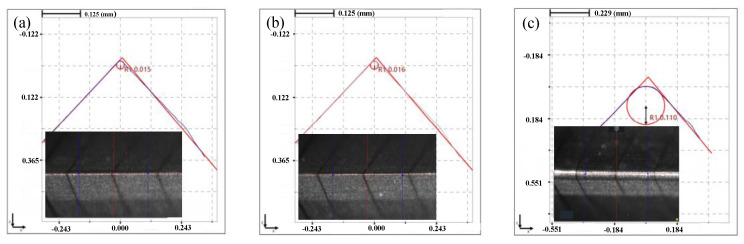
The edge shape diagram under different methods for 5 min: (**a**) initial, radius 15 μm. (**b**) STP, radius 16 μm. (**c**) FF-STP, radius 110 μm.

**Figure 6 micromachines-13-01631-f006:**
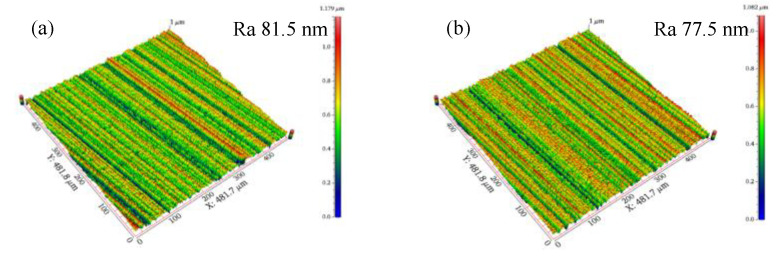
Topography of flank face: (**a**) STP. (**b**) FF-STP.

**Figure 7 micromachines-13-01631-f007:**
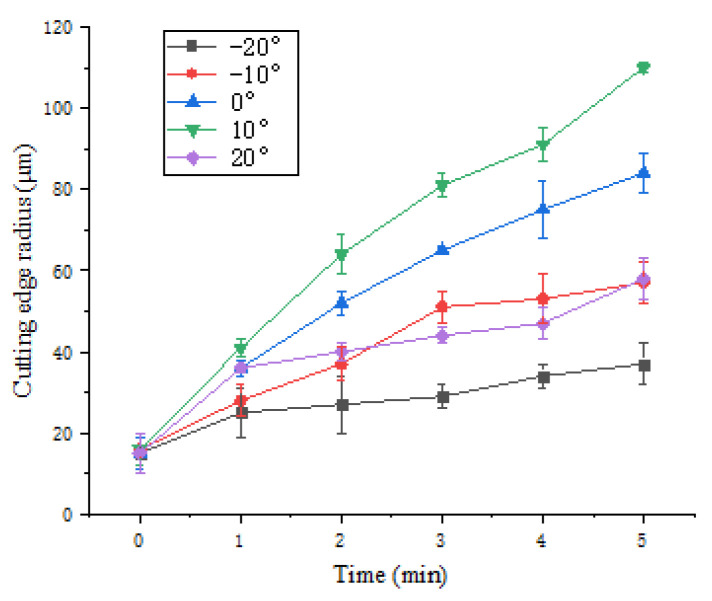
Experimental results under different polishing angles.

**Figure 8 micromachines-13-01631-f008:**
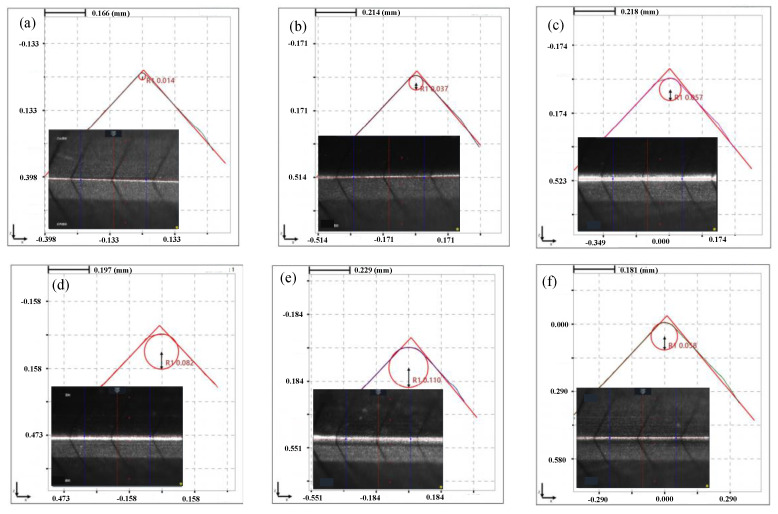
Edge shape diagram under different polishing angles for 5 min: (**a**) before preparation, (**b**) α = −20°, (**c**) α = −10°, (**d**) α = −0°, (**e**) α = 10°, (**f**) α = 20°.

**Figure 9 micromachines-13-01631-f009:**
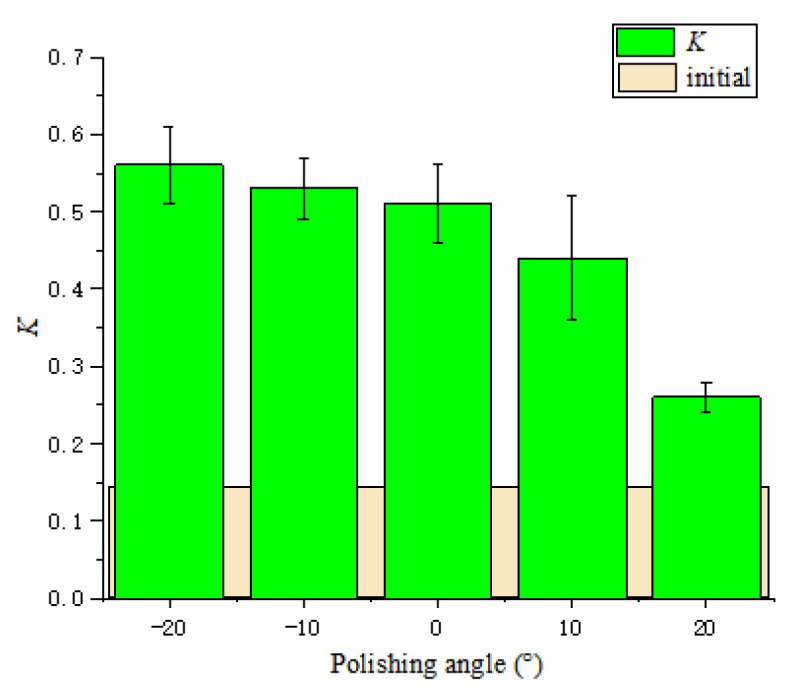
The change of the shape factor *K* with the polishing angle.

**Figure 10 micromachines-13-01631-f010:**
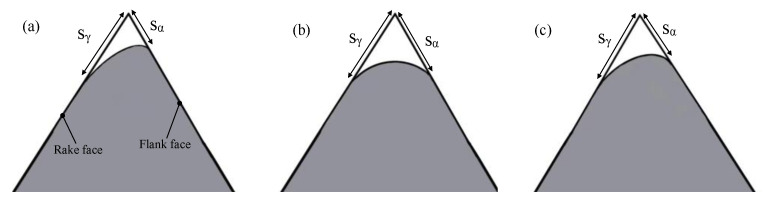
Schematic diagram of the change of *K*-factor: (**a**) initial, *K* = *s_α_/s_γ_* = 0.14, (**b**) −20°, *K* = *s_α_/s_γ_* = 0.56, (**c**) 20°, *K* = *s_α_/s_γ_* = 0.26.

**Figure 11 micromachines-13-01631-f011:**
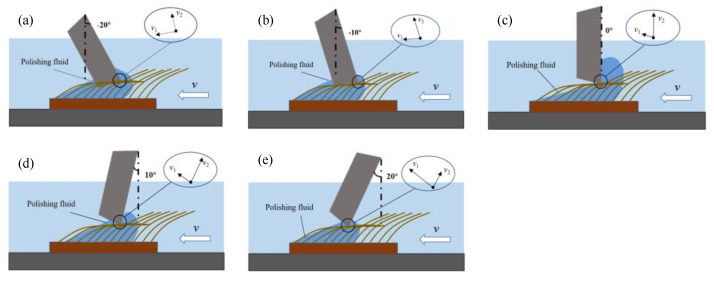
Schematic diagram of cutting edge preparation at different polishing angles: (**a**) –20°, (**b**) −10°, (**c**) 0°, (**d**) 10°, (**e**) 20°.

**Figure 12 micromachines-13-01631-f012:**
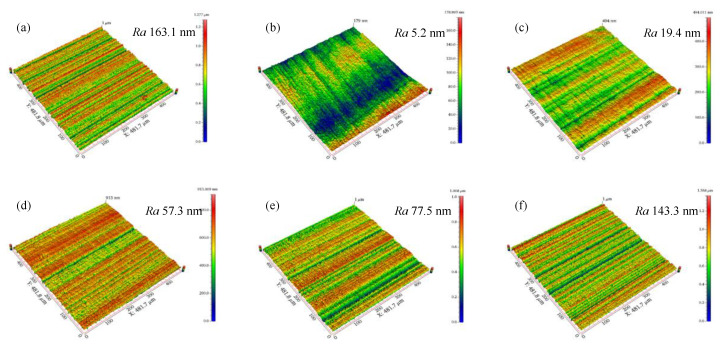
Micrographs and 3D micro-topography images of flank face. (**a**) initial, *Ra* 163.1 nm, (**b**) −20°, *Ra* 5.2 nm, (**c**) −10°, *Ra* 19.4 nm, (**d**) −0, *Ra* 57.3 nm, (**e**) 10°, *Ra* 77.5 nm, (**f**) 20°, *Ra* 143.3 nm.

**Figure 13 micromachines-13-01631-f013:**
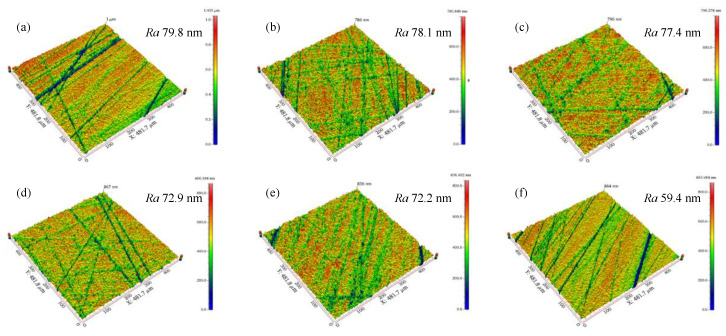
Micrographs and 3D micro-topography images of rake face. (**a**) initial, *Ra* 79.8 nm, (**b**) −20°, *Ra* 78.1 nm, (**c**) −10°, *Ra* 77.4 nm, (**d**) −0, *Ra* 72.9 nm, (**e**) 10°, *Ra* 72.2 nm, (**f**) 20°, *Ra* 59.4 nm.

**Figure 14 micromachines-13-01631-f014:**
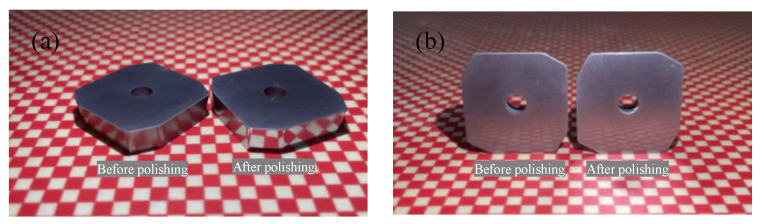
Comparison of cemented carbide insert surface quality before and after polishing: (**a**) flank face, (**b**) rake face.

**Figure 15 micromachines-13-01631-f015:**
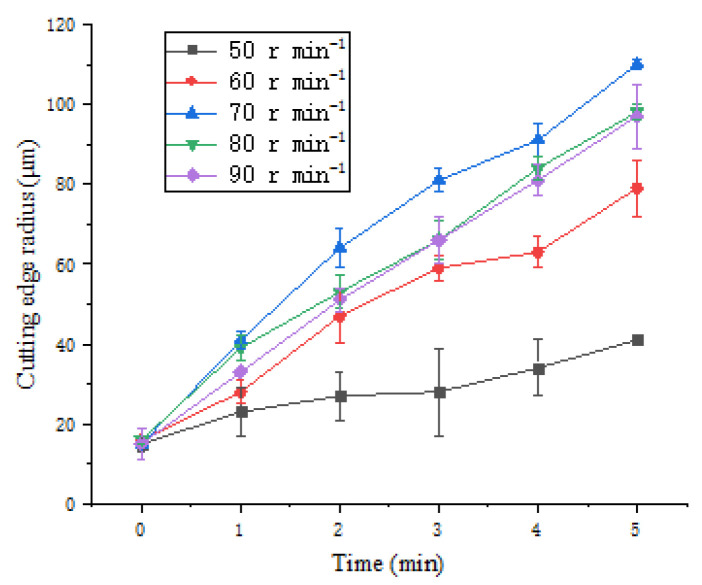
Experimental results under different polishing velocities.

**Figure 16 micromachines-13-01631-f016:**
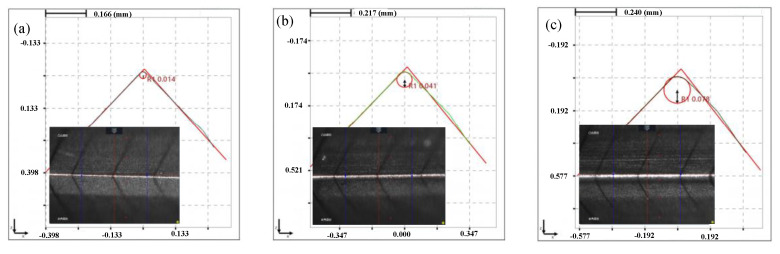
Edge shape diagram under different polishing velocity: (**a**) before preparation (**b**) *v* = 50 r·min^−1^, (**c**) *v* = 60 r·min^−1^, (**d**) *v* = 70 r·min^−1^, (**e**) *v* = 80 r·min^−1^, (**f**) *v* = 90 r·min^−1^.

**Figure 17 micromachines-13-01631-f017:**
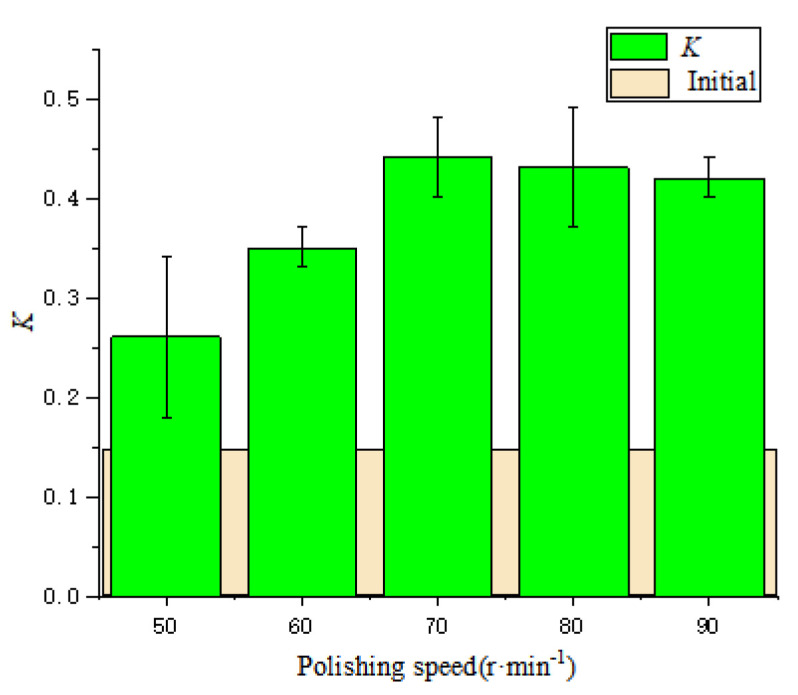
The change of the shape factor *K* with the polishing speed.

**Figure 18 micromachines-13-01631-f018:**
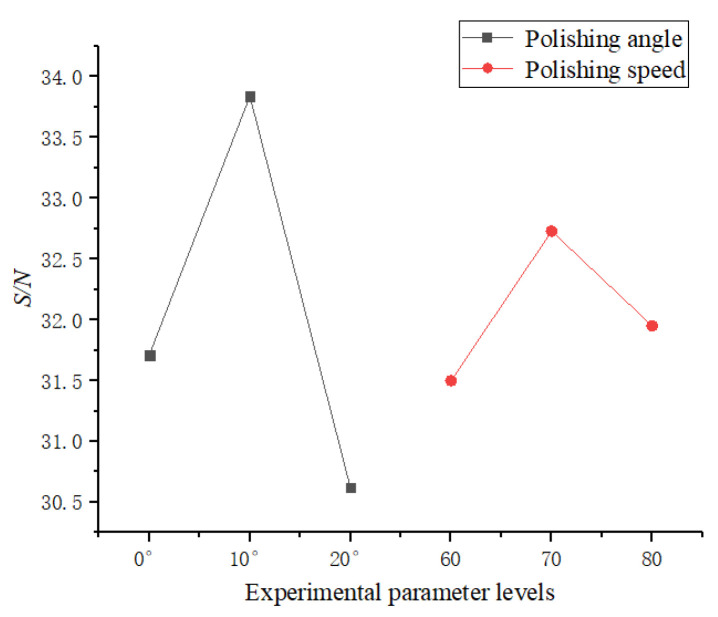
Plots of *S/N* ratio of each parameter level of *R*.

**Figure 19 micromachines-13-01631-f019:**
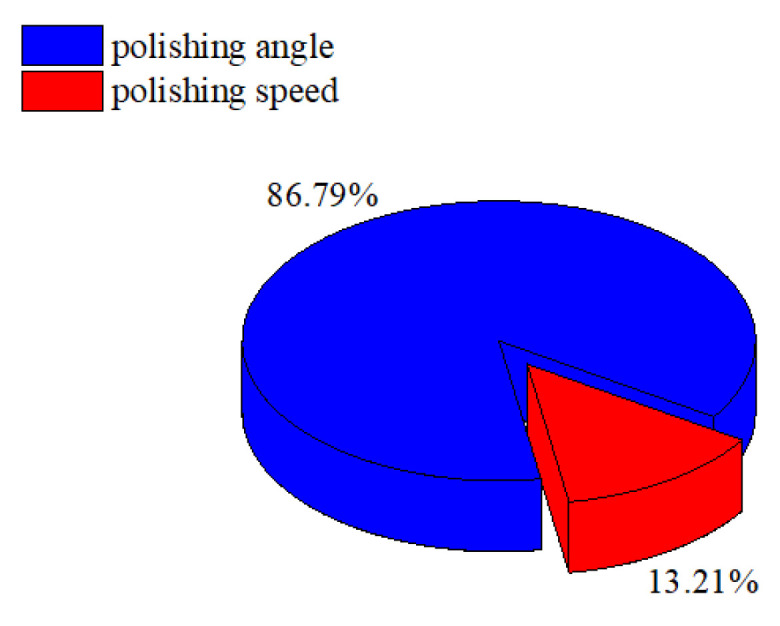
ANOVA results of *R*.

**Figure 20 micromachines-13-01631-f020:**
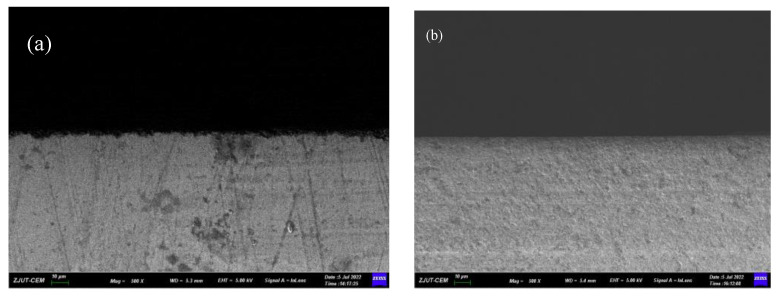
SEM images of cutting edge. (**a**) before the preparation, (**b**) after the preparation.

**Figure 21 micromachines-13-01631-f021:**
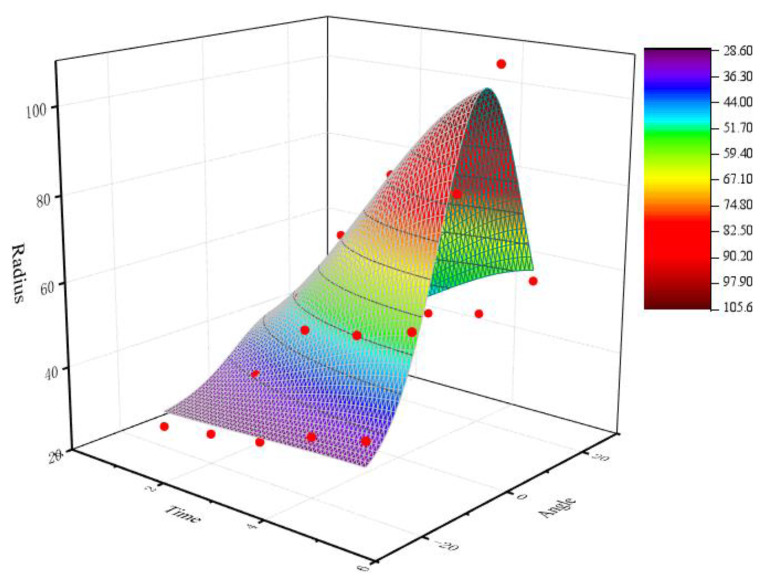
Fitting surface diagram of model 1.

**Figure 22 micromachines-13-01631-f022:**
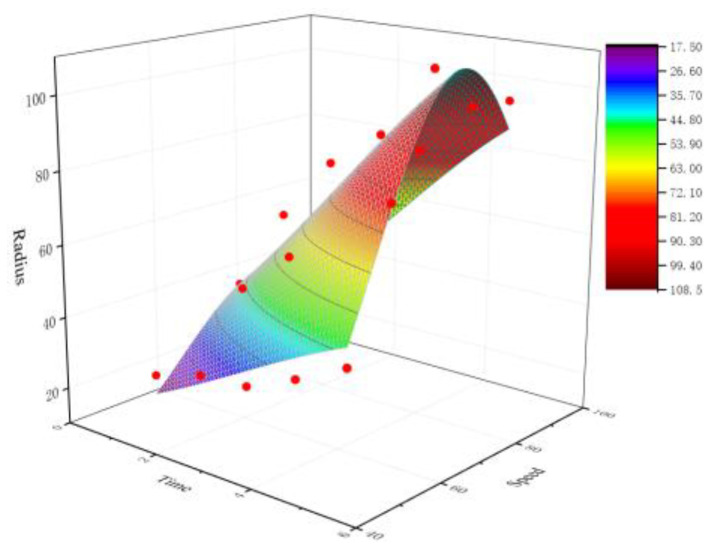
Fitting surface diagram of model 2.

**Table 1 micromachines-13-01631-t001:** Mechanical property of cemented carbide insert.

Material	Density (g/cm^3^)	Flexural Strength (MPa)	Hardness (HRA)	Fracture Toughness (MPa m^1/2^)
YG8	14.7	1500	89	2.5

**Table 2 micromachines-13-01631-t002:** Experimental conditions of experiment.

Parameters	Values	
	Group 1	Group 2
Polishing angle (°)	−20, −10, 0, 10, 20	10
Polishing speed (r·min^−1^)	70	50, 60, 70, 80, 90
Processing time per trial (min)	1	
Abrasive	Diamond, 5000#, 6 wt.%	

**Table 3 micromachines-13-01631-t003:** L9 (3^2^) Orthogonal experimental scheme.

Symbol	Factors	
Trial No.	A Polishing Angle (°)	B Polishing Speed (r min^−1^)
1	0	60
2	0	70
3	0	80
4	10	60
5	10	70
6	10	80
7	20	60
8	20	70
9	20	80

**Table 4 micromachines-13-01631-t004:** Experimental results.

No.	A	B	*R* (μm)	(dB)
1	0	60	35	30.88
2	0	70	43	32.67
3	0	80	38	31.60
4	10	60	46	33.26
5	10	70	54	34.64
6	10	80	48	33.62
7	20	60	33	30.37
8	20	70	35	30.88
9	20	80	34	30.63

**Table 5 micromachines-13-01631-t005:** Significance analysis of model 1.

*R*-Square *R*^2^: 0.93815				
	DOF	Sum of Square	Mean Square	*F*-Measure
Regression Coefficient	6	78,835.0221	13,139.17035	343.40004
Residual Error	19	726.9779	38.26199	

**Table 6 micromachines-13-01631-t006:** Significance analysis of model 2.

*R*-Square *R*^2^: 0.94947				
	DOF	Sum of Square	Mean Square	*F*-Measure
Regression Coefficient	6	102,858.13579	17,143.02263	414.47037
Residual Error	19	785.86421	41.36127	

## Data Availability

All data included in this study are available upon request from the corresponding author.

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
