# Peer review of "Preparation of Cemented Carbide Insert Cutting Edge by Flexible Fiber-Assisted Shear Thickening Polishing Method"

_micromachines, 2022, doi:10.3390/mi13101631_

Round 1
Reviewer 1 Report
FFSTP method is proposed for the preparatoin of the cutting edge of the cemented carbide insert, and its effectiveness has been validated by the experimental results in this manuscript. It should be useful for the practical production of the cutting tool.
Before publishing, I would like the authors to address the following concerns.
1.Fig. 2 and 7 can be enlarged to make it more clear.
2.As for the experiment of group 3, why chose 0,10,20deg?Why not -10,0,10 or -20,-10,0? Please explain.
3.The detail composition of the polishing slurry should be provided, since it has significant effect to the polishing process, such as the brand and type of the abrasive particle, non-newtonian fluid and additives.
4.The detail information of the fiber should also be provided, such as material, diameter, length, etc.
5.Subtitle of fig. 8 is missing.
6.As for the surface roughness measurement, measurement is conducted on which face is not provided. And how many points and their corresponding positions are also not clear.
7.The resolution of some figures are too low. Please improve.
Author Response
Please see the attachment.
Dear Editor and Reviewers:
Thank you for your concerning our manuscript entitled “Preparation of Cemented Carbide Insert Cutting Edge by Flexible Fiber-assisted Shear Thickening Polishing Method” (ID: 1866250). Those comments are all valuable and very helpful for revising and improving our paper, as well as the important guiding significance to our researches. We have studied your comments carefully and have made the recommended corrections that we hope to meet with approval. Revised parts are marked in different colors in the revised manuscript. The main corrections and the responses to reviewer’s comments are as follows:
Note:
Revisions are highlighted in different colors in the revised manuscript according to reviewers’ comments.
Responds to the reviewer’s comments (Text in italic style: the revised part of the manuscript.) :
Point 1: Fig. 2 and 7 can be enlarged to make it more clear.
Response 1: Fig. 2, Fig.7 and other pictures have been enlarged appropriately, so it can be seen more clearly.
Point 2: As for the experiment of group 3, why chose 0,10,20deg?Why not -10,0,10 or -20,-10,0? Please explain.
Response 2: Group 3 is based on the experiment of Group 1. It can be seen from Group 1 that the cutting edge preparation efficiency is higher at the angles of 0, 10, and 20, so these datas in Group 1 are selected for the Group 3 experiment.
Point 3: The detail composition of the polishing slurry should be provided, since it has significant effect to the polishing process, such as the brand and type of the abrasive particle, non-newtonian fluid and additives.
Response 3: The detail composition of the polishing slurry has been provided in the revised manuscript. ‘The polishing slurry was prepared by micro polyhydroxy polymer powders(Shandong, Hengren), deionized water, diamond particles (Zhengzhou Institute of Abrasives and Grinding), preservative (Casson) and dispersant (Ningbo Risheng New Material Co., Ltd.)’
This parts are marked in green background in section 3.2 of the revised manuscript.
Point 4: The detail information of the fiber should also be provided, such as material, diameter, length, etc.
Response 4: The detail information of the fiber has been provided in Section 3.2 of the revised manuscript. ‘and the flexible fiber tool used in the experiment is made of pig bristle with length 30 mm, diameter 200-250 μm and density 200–250 pieces/cm2.’
This parts are marked in Red in section 3.2 of the revised manuscript.
Point 5: Subtitle of fig. 8 is missing.
Response 5: The subtitle of Fig.8 have been added.
The revised parts are marked in Blue in the revised manuscript.
Point 6: As for the surface roughness measurement, measurement is conducted on which face is not provided. And how many points and their corresponding positions are also not clear.
Response 6: The sentence ‘and the surface roughness was measured by the white light interferometer (SuperView W1 Chotest)’ have been deleted and modified this sentence to ‘The roughness of three different positions on the machined surface were measured by the white light interferometer (SuperView W1 Chotest). The three measurement results are averaged, and the positions of the measurement points are shown in Figure 3(c)’. Besides, Figure 3(c) has been added to better explain the measured position.
The revised parts are marked in the yellow background in the last paragraph of section 3.2.
Point 7: The resolution of some figures are too low. Please improve.
Response 7: The low-resolution images have been adjusted in the revised manuscrip.
We appreciate for Editors/Reviewers’ warm work earnestly and hope that the correction will meet with approval. Once again, thank you very much for your comments and suggestions.
Yours sincerely,
Binghai Lyu

Reviewer 2 Report
This article is very practical and can be very helpful for cutting tools edge preparation.
The following improvements can further increase the article reading:
1-Page 3, section 3.1. The polishing angle is not clearly understandable while looking at figure 2b. Please consider representation this angle on Figure 1 too or on figure similar as Figure 11. The authors could also refer to Fig 11.
2- Figure 14. Comparison of cemented carbide inserts before and after polishing: The photos presented do not show the change on the cutting edge after polishing. Please consider improving the photos or the magnification to focus better on the edge geometry change.
3- Is there any reason why the authors have not considered the interaction between the polishing speed and the tilting angle on the responses? Such interaction could affect the selection of polishing angle and speed. Please elaborate.
Author Response
Please see the attachment.
Dear Editor and Reviewers:
Thank you for your concerning our manuscript entitled “Preparation of Cemented Carbide Insert Cutting Edge by Flexible Fiber-assisted Shear Thickening Polishing Method” (ID: 1866250). Those comments are all valuable and very helpful for revising and improving our paper, as well as the important guiding significance to our researches. We have studied your comments carefully and have made the recommended corrections that we hope to meet with approval. Revised parts are marked in different colors in the revised manuscript. The main corrections and the responses to reviewer’s comments are as follows:
Point 1: Page 3, section 3.1. The polishing angle is not clearly understandable while looking at figure 2b. Please consider representation this angle on Figure 1 too or on figure similar as Figure 11. The authors could also refer to Fig 11.
Response 1: The polishing angle on Figure 1 has been modified according to Figure 11.
Point 2: Figure 14. Comparison of cemented carbide inserts before and after polishing: The photos presented do not show the change on the cutting edge after polishing. Please consider improving the photos or the magnification to focus better on the edge geometry change.
Response 2: The interpretation of Figure 14 ‘Comparison of cemented carbide inserts before and after polishing’ may be misleading. Actually Figure 14 shows the change of the surface quality on rake face and flank face.The change on the cutting edge after polishing showed in Fig 8. So, it changes to ‘Figure 14. Comparison of cemented carbide inserts surface quality before and after polishing: (a) flank face. (b) rake face.’
This parts are marked in green in the revised manuscript.
Point 3: Is there any reason why the authors have not considered the interaction between the polishing speed and the tilting angle on the responses? Such interaction could affect the selection of polishing angle and speed. Please elaborate.
Response 3: In section 4.4, orthogonal experiments of polishing angle and speed were carried out by Taguchi method experiment to analyze the interaction relationship between polishing angle and polishing speed. So, a part of the content has been modified and added to the manuscript:
‘(1) In order to explore the degree of influence of the polishing angle and speed on the cut-ting edge radius and analyze the interaction between polishing angle and polishing speed, an orthogonal experiment based on Taguchi method was used.’
‘(2) The material removal of the cutting edge mainly depends on the relative velocity between polishing slurry and the cutting edge surface and the applied pressure. When the polishing angle is too large, the polishing slurry flows smoothly through the rake face with the flexible fiber, but the pressure on the cutting edge is reduced. When the polishing angle is too large, the polishing slurry almost impacts the edge of the insert, and the relative speed between the polishing slurry and the edge decreases. Changing the polishing speed in these two cases can hardly change the material removal rate efficiently at the cutting edge. In the first group of experiments, the angles 0, 10, 20 and the polishing speed 60 rmin-1, 70 rmin-1, 80 rmin-1 with higher preparation efficiency were selected.’
(3) With the increase of the polishing speed, the preparation efficiency first increases and then decreases. With the increase of the polishing angle, the preparation efficiency first increases and then decreases.
This parts are marked in grey background in section 4.4 of the revised manuscript.
We appreciate for Editors/Reviewers’ warm work earnestly and hope that the correction will meet with approval. Once again, thank you very much for your comments and suggestions.
Yours sincerely,
Binghai Lyu
